

# Intercomparison of tropopause height climatologies: High-Resolution radiosonde measurements versus ERA5 reanalysis

Yu Gou[1]; Jian Zhang[1*]; Wuke Wang[2]; Kaiming Huang[3]; Shaodong Zhang[3]

[1]Hubei Subsurface Multi–scale Imaging Key Laboratory, School of Geophysics and Geomatics, China University of Geosciences, Wuhan 430074, China

[2]School of environmental studies, China University of Geosciences, Wuhan 430074, China

[3]School of Electronic Information, Wuhan University, Wuhan 430072, China

*Correspondence to:* Dr. Jian Zhang (Email: zhangjian@cug.edu.cn)

**Abstract.** The tropopause plays a critical role in stratosphere-troposphere exchange and climate change. Its height is widely defined based on the World Meteorological Organization (WMO) threshold temperature gradient. High-resolution (5–10 m) soundings, therefore, are expected to substantially minimize uncertainties of tropopause height (TH) arising from limited vertical resolution and imprecise temperature measurements. The near-global coverage of high-resolution radiosonde data, accumulated from 2000 to 2023, offers valuable insights into climatological tropopause variability. While radiosonde observations are limited by spatiotemporal coverage, European Centre for Medium–Range Weather Forecasts Reanalysis v5 (ERA5) reanalysis datasets offer globally complete tropopause representations. To leverage both the high resolution of radiosonde measurements and the global coverage of ERA5, this study compares their tropopause height estimates and analyzes long-term trends across different latitude zones and seasons. The results indicate that the mean and absolute differences (radiosonde minus ERA5) in TH were 32 m and 336 m, respectively, with larger discrepancies observed during the spring season in the tropics (±20°). Overall, point-to-point comparisons (with strict spatio-temporal matching) indicate that ERA5 effectively captures climatological tropopause height variations in both time and space. Long-term trend analyses revealed increases of +5 m/year (radiosonde) and +3 m/year (ERA5) based on point-to-point comparisons. However, these site-specific trends may differ substantially from the long-term trends observed in ERA5 with complete spatiotemporal resolution, even showing opposite trends. Therefore, continued accumulation of high-resolution radiosonde profile data is crucial to further characterize tropopause changes in a warming climate.





### Introduction

The tropopause, marking the boundary between the turbulent troposphere and the stably stratified stratosphere, is a "gate" for exchange of energy, air masses, water vapor and so-called very short lived substances (Fueglistaler, 2009). Furthermore, given the impact of global warming and ozone depletion on the troposphere and stratosphere, tropopause variations can serve as an indicator of anthropogenic environmental influences (Santer et al., 2003). Moreover, its extreme sensitivity to climate variability and change makes the tropopause a pivotal factor in understanding and predicting future climate scenarios (Sausen and Santer, 2003; Seidel and Randel, 2006).

The tropopause exhibits unique chemical and dynamical characteristics, and its maintenance relies on complex interactions between large-scale and small-scale circulation patterns, deep convection, cloud formation, and radiation (Randel and Jensen, 2013). For instance, water vapor abundance can influence tropopause height (TH), as an increase in TH accompanies increased optical thickness to maintain a constant emitted temperature. In addition, TH is generally controlled by a combination of diabatic forcing and adiabatic dynamical effects (Zurita–Gotor and Vallis, 2013). These intricate mechanisms eventually lead to a marked difference in TH between the tropics and the poles, with the annual average reaching approximately 16 km in the tropics and 8 km in the polar regions. The subtropical jet streams (STJs), which are typically located at 20°–40° latitude in each hemisphere and 12 km height (Manney and Hegglin, 2018) will reinforces the Hadley circulation via "eddy pump" (Staten et al., 2018). The large-scale downwelling in the subtropical Hadley circulation sharply lowers the tropopause, sometimes creating a discontinuity known as the "subtropical tropopause break", which aligns with the STJ (Turhal et al 2024).

There are multiple ways to calculate the TH, relying on several empirical criteria based on properties exhibiting sharp transitions between the troposphere and stratosphere. The cold point tropopause (CPT) get the tropopause via the minimum temperature in the vertical temperature profile, which is unsuitable for extratropical regions (Highwood and Hoskins, 1998), and the dynamic tropopause (WMO, 1985; Hoinka, 1998), typically defined by potential vorticity (PV) thresholds of 1.5–4 potential vorticity unit (PVU) (Turhal et al., 2024), is less reliable in regions of low absolute potential vorticity, such as the tropics, and sometimes in mid-latitudes where strong anticyclonic flow prevails (Hoerling et al., 1991). However, the tropopause definition (WMO, 1957), which is thermodynamic, proposed by the World Meteorological Organization offers a more robust global approach (though it may occasionally fail in polar regions), providing reliable TH estimates from various datasets (Hoffmann and Spang, 2022).

Current primary data sources for TH determination include in-situ measurements from radiosonde, meteorological reanalysis datasets, and radio occultation data from the Constellation Observing System for Meteorology Ionosphere and Climate (COSMIC) Data Analysis and Archive Center (CDAAC) of the Global Navigation Satellite System (GNSS). Radiosonde data, collected at stations worldwide, feature long-term records with high accuracy and reliability, yet suffer from sparse and highly uneven global coverage. Global atmospheric reanalysis products provide comprehensive long-term atmospheric information through continuously improved forecast models, observational data, and assimilation



schemes (Fujiwara et al., 2017). Modern reanalysis datasets, such as ERA5 and ERA-Interim developed
by the European Centre for Medium-Range Weather Forecasts (ECMWF) (Dee et al., 2011), and
Modern-Era Retrospective analysis for Research and Applications, Version 2 (MERRA-2) produced by
National Aeronautics and Space Administration (NASA) Global Modeling and Assimilation Office
(GMAO) (Gelaro et al., 2017), overcome the spatial and temporal limitations of observational records,
offering a global perspective of the tropopause. GNSS -based datasets such as COSMIC provide high-
density measurements with near-global coverage, making them particularly suitable for analyzing global-
scale tropopause characteristics (Son et al., 2011).

More and more evidence suggests an upward trend in TH under a changing climate (Santer et al.,
2003; Sausen and Santer, 2003; Seidel and Randel, 2006; Añel et al., 2006). For instance, Seidel and
Randel (2006) observed a 64±21 m/decade upward trend from 1980 to 2004, and Son et al. (2009)
projected continued future increase, albeit with a weaker trend. Analyses by Xian and Homeyer (2019)
using radiosonde observations and reanalysis datasets from 1981–2015 detected a significant upward
trend (40–120 m/decade). Meny et al. (2021) reported a TH increase of 50–60 m/decade (2001–2020) in
the Northern Hemisphere based on radiosonde data. More recently, Zou et al. (2023) leveraged European
Centre for Medium–Range Weather Forecasts (ECMWF) Reanalysis v5 (ERA5) data to reveal a
widespread upward and cooling trend in the tropical tropopause from 1980–2021, demonstrating an
increase of approximately 60±10 m/decade (95% confidence). It is worthwhile to note that these
radiosonde-related studies use data from the Integrated Global Radiosonde Archive (IGRA), a global
dataset with coarse vertical resolution (approximately 300 to 400 m) (Durre et al., 2006, 2018), and the
ERA5 137-level model, however, has a vertical resolution of roughly 300 m at altitudes of 5 km and
above, both these radiosonde data and reanalysis/model datasets, while useful for studying global
tropopause variations, suffer from limitations in vertical resolutions (Raman and Chen, 2014).

In recent decades, newly developed high vertical resolution (5–10 m) of the radiosonde data allows
for fine detailed observation of temperature structure changes within the troposphere and stratosphere.
High-resolution sounding data benefits the detection of fine-scale tropopause structures research like
multiple tropopause events and provide more reliable estimates of TH changes across hundreds of
stations. Furthermore, the global dataset is accumulating, with increasingly long records approaching
climate-relevant timescales. Examples include the US (starting in 2005; Ko et al., 2019), China (starting
in 2011), and Europe (starting in approximately 1991).

However, reanalysis datasets offer a globally continuous spatial and temporal representation of TH,
complementing radiosonde data. Datasets like ERA-Interim are widely recognized for their utility in
climate monitoring (Dee et al., 2011), overcoming the spatial resolution limitations of radiosonde data.
Therefore, ERA5 has naturally been subjected to comparisons and evaluations with radiosonde in
different regions (e.g., Zhu et al., 2021; Velikou et al., 2022; Hoffmann and Spang., 2022). However,
global comparative analyses of high-resolution (10 m) radiosonde data with ERA5 at the tropopause level
remain scarce.



Consequently, this study addresses the following questions: (1) How does ERA5 perform in terms of obtaining TH climatologies compared to high-resolution detection? (2) What the global long-term change of TH according radiosonde data and ERA5 reanalysis data? To this end, section 2 details the data and the methods we used. Section 3 presents the comparative analysis and long-term change of TH

derived from radiosonde observations and ERA5 reanalysis data. Section 4 ends with a short summary.

**2. Data and method**

**2.1 Radiosonde and ERA5**

Radiosondes are fundamental and crucial data sources for numerical weather prediction models. They are typically carried aloft by weather balloons and burst at altitudes of approximately 27 km (Kumar,

2023). As the radiosonde ascends, it transmits meteorological data including temperature, pressure, relative humidity and air pressure to ground, sea, or air-based receiving stations (Durre et al., 2006). Globally, radiosondes are launched at approximately 800 sites regularly twice a day (Ingleby et al., 2016; Durre et al., 2018). Radiosonde data are widely used in studies of planetary boundary layer height (Sorbjan and Balsley, 2008; Seidel et al., 2010), tropopause structure (Birner, 2006; Seidel and Randel,

2006; Añel et al., 2008; Sunilkumar et al., 2017) and gravity waves (Ki and Chun, 2010; Yoo et al., 2018). Radiosonde data have advantages such as in-situ measurements, high vertical resolution, and generally reliable datasets. However, limitations include relatively low and uneven spatial resolution, as well as a limited temporal frequency of typically twice daily measurements.

ERA5 is the latest fifth-generation global atmospheric reanalysis product, stands out as one of the

best high–resolution atmospheric reanalysis products currently available, utilizing the ECMWF Integrated Forecasting System (IFS) Cy41r2, combined with a 4D-Var assimilation scheme (Hersbach et al., 2020). This remarkable initiative within the Copernicus Climate Change Service (Thépaut et al., 2018) benefits from advancements in modeling and data assimilation over a decade, providing a comprehensive and high-quality record of essential climate variables (Raoult et al., 2017).

Following Guo et al. (2021) and Zhang et al. (2022), we utilized a high-vertical-resolution radiosonde (HVRRS) dataset spanning 2000 to 2023 (24 years), compiled from multiple sources including the China Meteorological Administration (CMA), the National Oceanic and Atmospheric Administration (NOAA) of the United States, the German Deutscher Wetterdienst (Climate Data Center), the Centre for Environmental Data Analysis (CEDA) of the United Kingdom, the Global Climate

Observing System (GCOS) Reference Upper-Air Network (GRUAN), and the University of Wyoming.

The data with a vertical resolution of 5–10 meters, ultimately sampled at 10 meters by applying a cubic spline interpolation. Data acquisition was typically performed at 00 UTC and 12 UTC each day. To ensure data quality, only records with at least 10 days (at least one record per day) for which the World Meteorological Organization (WMO) definition of the first tropopause was derived were

considered to be valid month and radiosonde stations with at least 10 valid months per year were considered to be valid in that year. Two distinct data selection criteria were established according to





study objectives: (1) For radiosonde-ERA5 intercomparison, we analyzed 222 stations (≥5 valid years each) with 1,530,517 vertical profiles during years 2000–2023; (2) For long-term TH trend detection, we had 109 stations (≥10 valid years each) containing 1,103,730 vertical profiles (Fig. 1).

Hoffmann and Spang. (2022) employed the WMO definition to calculate global THs from ERA5 data, making this ERA5-based product available for research purposes. This study utilizes the 2000–2023 ERA5-based data, characterized by a horizontal resolution of $0.3\,° \times 0.3\,°$ and a temporal resolution of 1 hour.

### 2.2 WMO-defined tropopause

The WMO tropopause definition is more robust and generally applicable across a wider range of latitudes. Therefore, calculating TH from radiosonde data using the WMO definition is currently the most suitable method for global tropopause comparisons. As provided by the WMO:

"(a) The first tropopause is defined as the lowest level at which the lapse rate decreases to 2 ℃/km or less, provided also the average lapse rate between this level and all higher levels within 2 km does not

exceed 2 ℃/km;

(b) if above the first tropopause the average lapse rate between any level and all higher levels within 1 km exceed 3 ℃/km, then a second tropopause is defined by the same criterion as under (a). This tropopause may be either within or above the 1 km layer."

The lapse rates are calculated as follows:

$$\Gamma(z_i) = -\frac{\delta T}{\delta z} = -\frac{T_{i+1} - T_{i-1}}{z_{i+1} - z_{i-1}} \tag{1}$$

with T represents temperature, and z represents geopotential height.

Following the WMO definition, Figure 2 presents four examples of radiosonde temperature profiles with corresponding nearest ERA5 profiles. Figures 2a and 2b show that radiosonde and ERA5 have a matched temperature profile, yet reveal significant discrepancies in tropopause identification. While case

(a) shows good agreement, case (b) exhibits a distinct inversion layer detected by the high vertical-resolution radiosonde (resulting in a much lower TH than ERA5). Cases (c) and (d) illustrate more pronounced profile differences: in (c), the rightward shift of the ERA5 temperature profile yields similar TH despite large temperature differences, case (d) displays both the greatest profile dissimilarity and a radiosonde-derived tropopause lowered by an inversion layer. This highlights how fine-scale thermal

structures resolved by high vertical-resolution radiosondes may complicate tropopause detection. The existing WMO definition could be further refined.

### 3. Result

### 3.1 Radiosonde-ERA5 tropopause height comparison and its seasonal variation

Figure 3 demonstrates that the THs derived from the radiosonde observations and ERA5 reanalysis

data are in strong agreement. The linear regression equation (y = 0.96x + 0.55) derived from the kernel

density scatter plot (left panel) closely follows the 1:1 line. In the lower-left region of the kernel density scatter plot, the discrepancy between radiosonde and ERA5 is small, with the radiosonde-derived THs being slightly higher on average than those from ERA5. However, the situation becomes more complex in the upper-right region, where the divergence between radiosonde and ERA5 suddenly increases, and the consistency along the regression line begins to diminish. This might suggest that both datasets exhibit certain limitations in capturing the TH within the subtropical tropopause break region. Figure 3 (right panel) displays the distribution of differences between radiosonde and ERA5. On average, TH derived from radiosondes is 32 m higher than that from ERA5, with a mean absolute difference of 336 m. The root mean square error is 756 m, while the Pearson correlation coefficient reaches 0.958 (significant level < 0.05), indicating a statistically significant agreement. Overall, the discrepancy between ERA5 and radiosonde data is minimal, demonstrating strong consistency between the two datasets.

Figure 4 presents the global distribution of mean differences (radiosonde minus ERA5) and mean absolute differences between the two datasets. Across all stations, 85% (187 stations) showed higher mean THs in radiosonde data compared to ERA5. At 206 stations (93%), the mean absolute difference remains within 500 m. Table 1 displays the year-by-year comparative analysis from 2000 to 2023, revealing a gradual increase in observation data over time. While the overall discrepancy between radiosonde and ERA5 remains relatively stable, the mean difference (radiosonde minus ERA5) shows a significant decrease during 2007–2016. The underlying causes for this reduction remain unclear and require further investigation.

Figure 5 presents 24-year (2000–2023) seasonal mean TH data from radiosonde and ERA5 across different latitudinal bands. Radiosonde and ERA5 data were divided into seven climate zones: Northern Hemisphere/Southern Hemisphere polar (70 °–90 °), Northern Hemisphere/Southern Hemisphere mid-latitude (40 °–70 °), Northern Hemisphere/Southern Hemisphere subtropics (20 °–40 °), and tropics (20 ° S–20 ° N) (Houchi et al., 2010). Seasonal analysis was performed on the station data within each latitudinal band. Due to the uneven distribution of radiosonde stations, the number of stations varies significantly across latitude bands, as indicated on the right side of each latitudinal title in Figure 5. The 40 ° N–70 ° N band contains the most stations (109 stations), while the 70 ° S–90 ° S band has only two stations. Except in the tropics and Southern Hemisphere polar region, the seasonal TH variations show consistent patterns: TH reaches maximum values during autumn (June–July–August) and winter (September–October–November), and minimum values during spring (December–January–February) and summer (March–April–May). This pattern is observed in all regions except the tropics and the Southern Hemisphere polar region.

### 3.2 Tropopause height long-term trend derived from Radiosonde and ERA5

For long-term trend analysis, we used data with at least ten years of valid records. Figure 6 presents the annual mean trends over a 24-year period from ERA5 globally. In Figure 6a, the long-term trend of the TH derived from ERA5 data shows a strong latitudinal dependence. Approximately 75% of the regions exhibit a positive trend, with a global average increase of 4 m/year. In contrast to the latitudinal



distribution of TH—which is higher at low latitudes and lower at high latitudes—the long-term trend generally displays an opposite pattern: lower trends at low latitudes and higher trends at high latitudes. Notably, a significant decreasing trend is observed along the 30 ° S band in tropical regions. Figure 6b compares the long-term trends of the TH between radiosonde and ERA5. The results indicate that 81% of the radiosonde stations show higher trends than ERA5. Specifically, the global TH change is +5 m/year based on radiosonde data.

Figure 7 displays the annual variations of TH across different latitudinal bands from 2000 to 2023, as observed by radiosonde and ERA5. The results show strong agreement between radiosonde and ERA5 in terms of the annual mean TH. However, by analyzing the regression lines of annual variations in each latitudinal band (Figure 7) along with Table 2, we find significant differences in the long-term trends of TH across latitudes. The long-term trend values for TH derived from radiosonde observations and spatially/temporally collocated ERA5 data show high consistency across latitudinal bands. Nevertheless, due to the limited number of stations, caution should be exercised when interpreting the trends in the following regions: 70 ° S–90 ° S, 20 ° S–40 ° S, and 70 ° N–90 ° N. The global mean of ERA5 TH trends, averaged over the 2000-2023 period and categorized by latitudinal bands, shows close alignment with radiosonde observations in terms of global variability. However, notable discrepancies exist in specific latitudinal bands. For instance, in the 40 ° N–70 ° N band, the ERA5 reanalysis shows a deviation of -27 m/year compared to the radiosonde data.

## 4. Conclusion and discussion

TH is an indispensable metric in climate change research, directly influenced by the temperature structure of the troposphere and stratosphere. Direct observations of the tropopause via radiosonde are generally most reliable, but are limited by the uneven global distribution of measurement stations, posing a significant constraint on global climate change studies. While radiosondes represent the most reliable data source, they suffer from stringent spatiotemporal limitations. In contrast, the ERA5 reanalysis dataset provides complete spatiotemporal coverage and has been widely utilized in atmospheric studies. Therefore, a systematic intercomparison between high-resolution radiosonde and ERA5 THs is essential to both reconcile dataset discrepancies and optimize their combined use for robust trend analysis.

This study investigates the spatial and temporal discrepancies between ERA5-derived THs and high-resolution radiosonde data. Building upon previous research by Hoffmann and Spang (2022), our investigation confirms an overestimation (32 m) of TH in radiosonde compared to ERA5. And WMO-defined thermodynamic tropopause may share mathematical similarities with thin, low-altitude temperature inversions detected in high-resolution radiosonde profiles, introducing complexity in TH determination. In the statistical analysis, the differences between radiosonde and ERA5 are more pronounced in the subtropical region. This discrepancy may be linked to the subtropical tropopause break phenomenon. Regarding the seasonal variation of TH, except in the tropics and the Southern Hemisphere polar region, TH peaks in autumn and winter while being lower in spring and summer (seasons are



reversed between hemispheres). However, radiosonde and ERA5 exhibit strong consistency in TH, with only minor differences about 32 m.

Comprehensive validation against high-resolution radiosonde observations demonstrates ERA5's exceptional performance in capturing TH characteristics, including absolute values, temporal variations, and spatial correlations. Application of the WMO tropopause definition to high-resolution soundings reveals potential limitations, particularly in subtropical regions where strong temperature inversions may lead to detection artifacts. These thin but intense inversion layers are often unresolved by coarse-resolution temperature profiles.

The global tropopause elevation trend is quantified at 50 m/decade based on radiosonde measurements. Given the 10-m vertical resolution of the sounding data, this rising trend represents a statistically robust signal, while ERA5 shows a trend of +3 m/year, with 81% of radiosonde stations exhibiting higher trends than ERA5. Point-to-point comparisons indicate ERA5 systematically underestimates long-term TH trends by approximately 40% relative to radiosonde benchmarks. This discrepancy warrants further investigation through coordinated model-observation intercomparison studies. The ERA5-based global data analysis reveals that 75% of regions exhibit an increasing trend in TH, and the long-term trend shows a latitudinal dependence: generally weaker in low latitudes and stronger in high latitudes, decreasing from +13 m/year to 0 m/year. Notably, the 20 °S–40 °S latitudinal band is the only zone displaying a significant decreasing trend (−6 m/year) in TH.

However, the variation of tropopause height with climate, according to ERA5, exhibits significant latitudinal and longitudinal differences. But many regions lack high-resolution, continuous radiosonde observations for cross-validation. This necessitates further verification through expanded station coverage and longer-term observations, particularly in the subtropical regions and polar zones of the Southern Hemisphere.

**Acknowledgement**

The authors would like to acknowledge the National Meteorological Information Centre (NMIC) of CMA, NOAA, the Deutscher Wetterdienst (Climate Data Center), the UK Centre for Environmental Data Analysis (CEDA), GRUAN, ECMWF, and the University of Wyoming for continuously collecting and generously providing high-resolution radiosonde data.

**Financial support**

This study jointly supported by the National Natural Science Foundation of China under grants 42205074.

**Competing interests**

The contact author has declared that neither they nor their co–authors have any competing interests.

**Data availability**

The authors would like to acknowledge the National Meteorological Information Centre (NMIC) of CMA, NOAA, German Deutscher Wetterdienst (Climate Data Center), UK Centre for Environmental



Data Analysis (CEDA), GRUAN, and the University of Wyoming (https://catalogue.ceda.ac.uk/, CMA, 2025; https://www.aparc-climate.org/data-centre/data-access/us-radiosonde/, NOAA, 2025; https://opendata.dwd.de/climate_environment/CDC/observations_germany/radiosondes/high_resolutio n/historical/, Deutscher Wetterdienst, 2025; http://data.cma.cn/en, CEDA, 2025; https://www.gruan.org/data/file-archive/rs92-gdp2-at-lc/, GRUAN, 2025; http://weather.uwyo.edu, The University of Wyoming, 2025) for providing the high-resolution sounding data. And the ERA5 reanalysis dataset can be accessed at https://datapub.fz-juelich.de/slcs/tropopause/ (Hoffmann and Spang, 2022).

**Author contributions**

JZ conceptualized this study. YG carried out the analysis with comments from other co–authors. YG and JZ wrote the original manuscript. WW, SZ provided useful suggestions for the study. All authors contributed to the improvement of paper.

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

**Tables and Figures**

**Table 1.** Statistical analysis by years, RS-E5: Radiosonde minus ERA5-based, Bias: average of differences (ERA5-based minus Radiosonde), Abs-bias: average of absolute values of differences, RMSE: root mean square error, R*: Pearson correlation coefficient ($p < 0.05$).

| Year | Valid record | Bias(RS-E5) (m) | Abs-bias(RS-E5) (m) | RMSE (m) | R* |
|------|--------------|-----------------|---------------------|----------|------|
| 2000 | 4651 | 44 | 243 | 460 | 0.938 |
| 2001 | 7120 | 43 | 236 | 446 | 0.957 |
| 2002 | 7335 | 56 | 228 | 376 | 0.969 |
| 2003 | 7377 | 40 | 231 | 417 | 0.962 |
| 2004 | 7739 | 48 | 227 | 401 | 0.964 |
| 2005 | 8105 | 35 | 267 | 527 | 0.943 |
| 2006 | 11259 | 35 | 302 | 632 | 0.960 |
| 2007 | 25650 | 14 | 358 | 781 | 0.951 |
| 2008 | 40154 | 20 | 362 | 795 | 0.953 |
| 2009 | 47716 | -28 | 422 | 939 | 0.929 |
| 2010 | 54066 | 20 | 347 | 764 | 0.960 |
| 2011 | 72234 | -12 | 399 | 918 | 0.938 |
| 2012 | 71474 | -20 | 387 | 906 | 0.941 |
| 2013 | 71930 | -1 | 375 | 885 | 0.945 |
| 2014 | 78010 | 22 | 366 | 837 | 0.952 |
| 2015 | 87345 | 28 | 350 | 771 | 0.959 |
| 2016 | 80738 | 21 | 381 | 861 | 0.952 |



| 2017 | 89145 | 51 | 336 | 732 | 0.962 |
|------|-------|----|-----|-----|-------|
| 2018 | 153446 | 46 | 312 | 685 | 0.966 |
| 2019 | 126090 | 57 | 286 | 621 | 0.972 |
| 2020 | 120414 | 58 | 285 | 612 | 0.972 |
| 2021 | 103729 | 45 | 375 | 852 | 0.948 |
| 2022 | 110719 | 51 | 304 | 662 | 0.966 |
| 2023 | 144071 | 47 | 304 | 692 | 0.964 |
| Total | 1530517 | 32 | 336 | 756 | 0.958 |

**Table 2.** Tropopause height trend from different latitude zones (Station data are absent for the 40 °S–70 °S region). ERA5-P is a one-dimensional data that corresponds one-to-one in time and space with the Radiosonde data, and ERA5-F is the overall mean of the ERA5 global trend averages over the period 2000 to 2023 in a two-dimensional plane divided according to latitudinal bands.

| Latitudinal zone | Number of stations | Radiosonde (m/year) | ERA5-P (m/year) | ERA5-F (m/year) |
|------------------|--------------------|---------------------|-----------------|-----------------|
| 90 °N–70 °N | 3 | 5 | 3 | 10 |
| 40 °N–70 °N | 56 | 33 | 32 | 6 |
| 20 °N–40 °N | 37 | -1 | -5 | 4 |
| 20 °S–20 °N | 8 | 11 | 8 | 0 |
| 20 °S–40 °S | 2 | -8 | -10 | -6 |
| 40 °S–70 °S | \ | \ | \ | 3 |
| 70 °S–90 °S | 2 | -34 | -32 | 13 |
| Global | 108 | 5 | 3 | 4 |






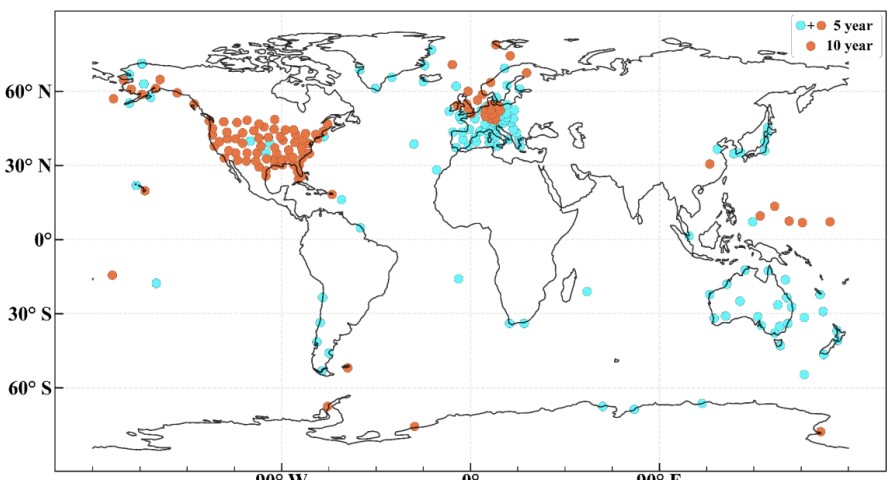

**Figure 1.** Spatial distribution of the qualifying radiosonde stations used in this study. The number of qualifying radiosonde stations with valid data records for 10 years (indicated by blue and orange dot) is 108, and that for 5 years (indicated by orange dot) is 222. The United States and Europe are data-intensive.


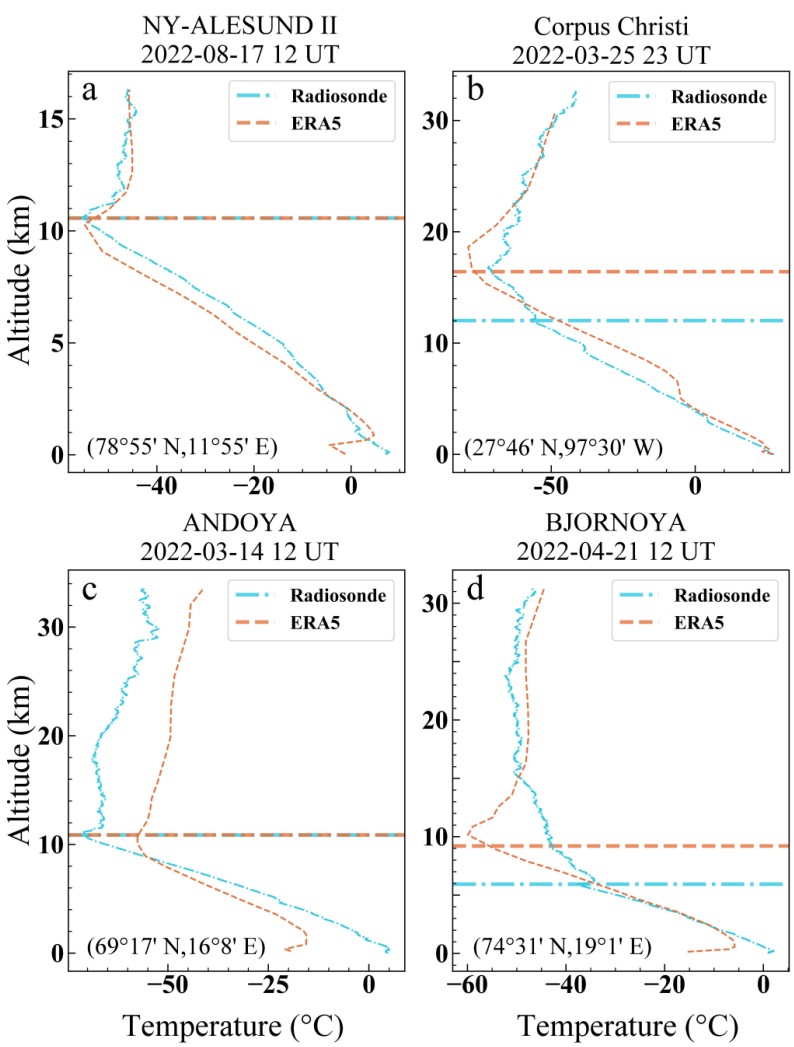

**Figure 2.** Cases of tropopause height by Radiosonde and ERA5 temperature profile. Orange repent ERA5, cyan
repent Radiosonde and the horizontal line represents the tropopause. Radiosonde location, date and hour are
marked in each subplot.



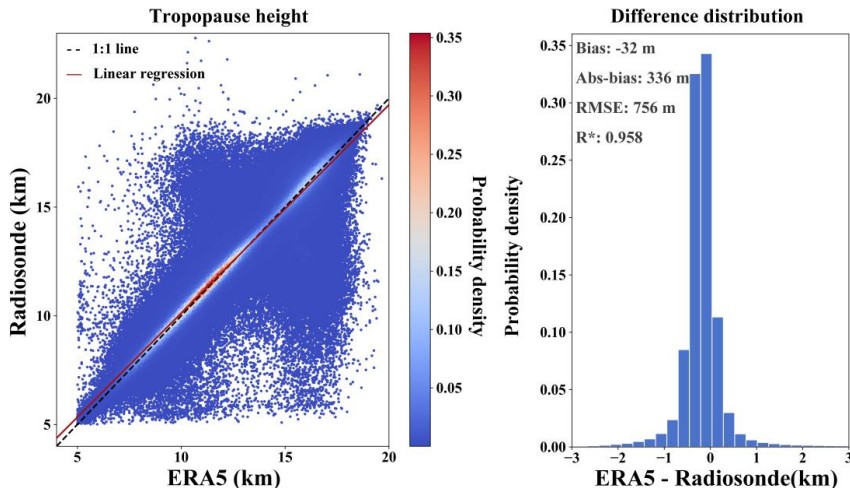


**Figure 3.** Kernel density scatter plot (left) and probability density histogram (right) comparing tropopause heights. The black dotted line indicates a 1:1 relationship, the red line shows the linear regression (y = 0.96x + 0.55). Bias (average of differences, ERA5 minus Radiosonde), absolute bias (average of absolute values of differences), RMSE (root mean square error), and R (Pearson correlation coefficient, p < 0.05) are also indicated.


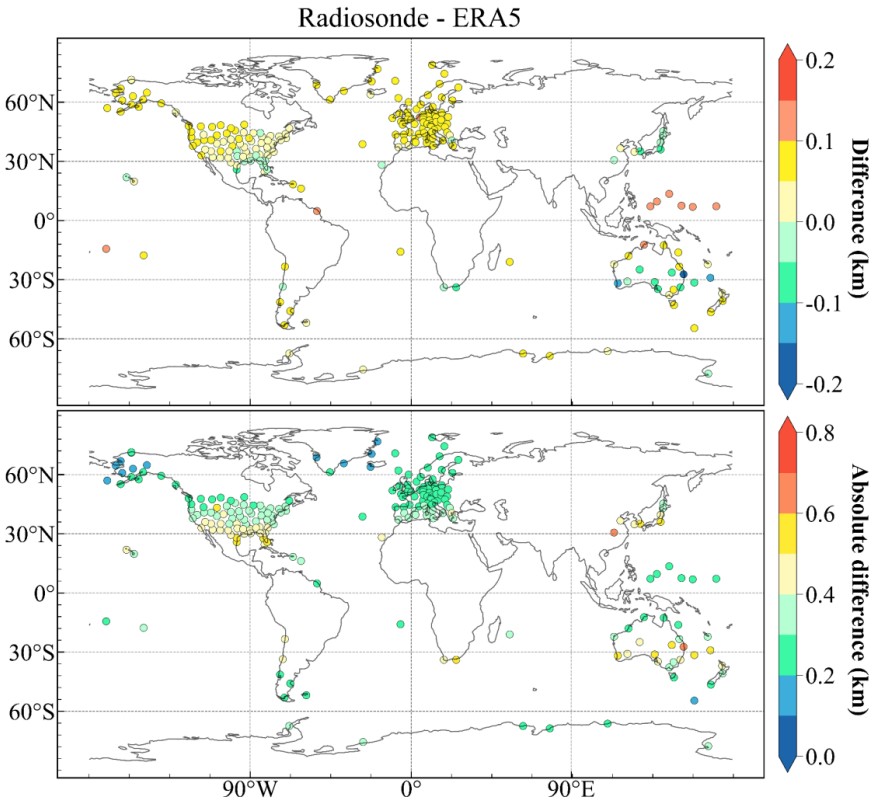

**Figure 4.** Global distribution of station-mean differences between radiosonde observations and ERA5 (upper panel) and station-mean absolute differences (lower panel).


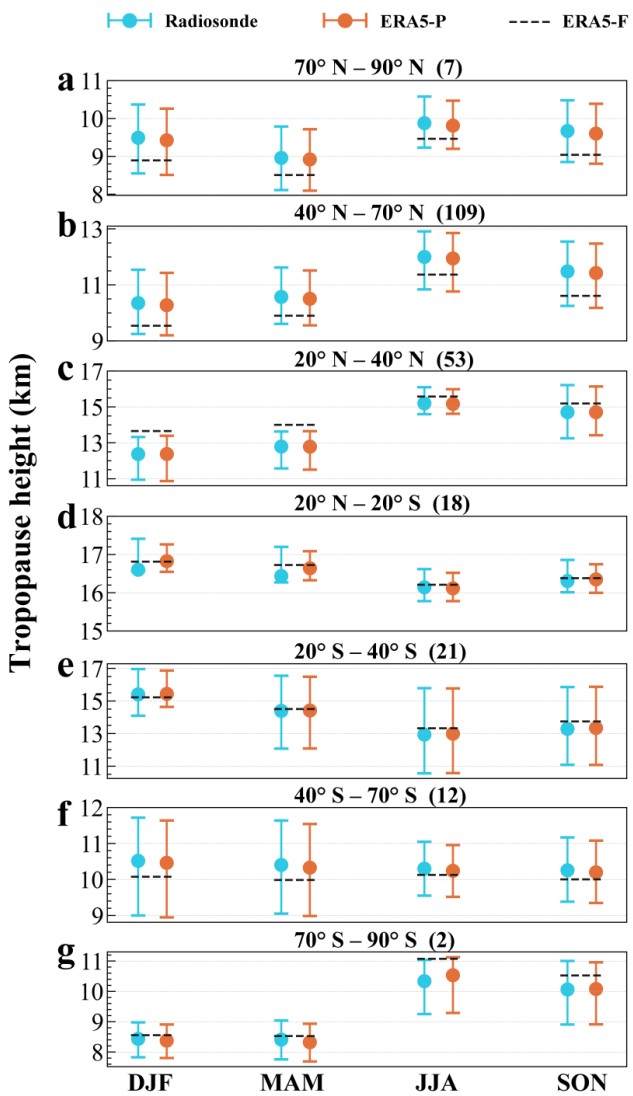

**Figure 5.** Seasonal change across 7 latitudinal bands. The number of stations on the right side of the latitude band titles. ERA5-P is a one-dimensional data that corresponds one-to-one in time and space with the radiosonde data, and ERA5-F is the overall mean of the ERA5 global seasonal averages over the period 2000 to 2023 in a two-dimensional plane divided according to latitudinal bands. DJF: December–January–February, MAM: March–April–May, JJA: June–July–August, SON: September–October–November. The dots represent the mean, and the bar represents the 25% or 75% quantiles.




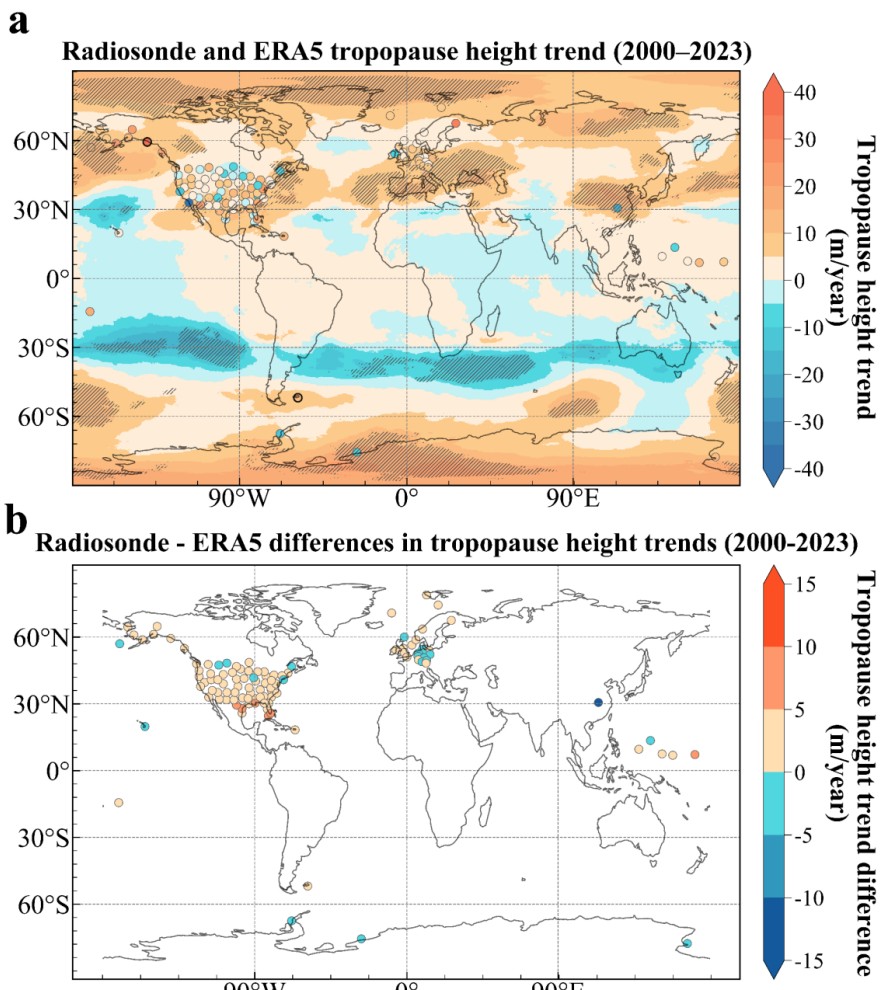

**Figure 6.** Global map of tropopause height trend (**a**) and its difference (**b**) using radiosonde and ERA5 data. The shaded area (black thick ring) in (**a**) indicates that the non-parametric Mann-Kendall test was used to examine the statistical significance of the trend observed on ERA5 (radiosonde) data at a significance level of 5%.



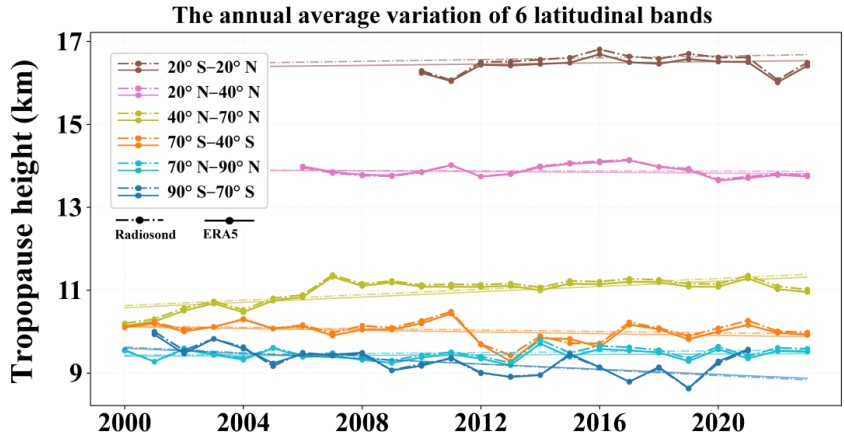


**Figure 7.** Annual variation (2000–2023) of radiosonde and ERA5 in six latitude bands (no data available for 40 °
S–70 °S). The dashed line represents radiosonde and the solid line represents ERA5.