# Peer review of "Intercomparison of tropopause height climatologies: High-Resolution radiosonde measurements versus ERA5 reanalysis"

_EGUsphere, 2025_

## Author Comment (AC1)

**Response to Referee**

We thank the reviewer for your time and effort in providing such insightful and constructive feedback.

We have carefully addressed all the concerns raised. The suggestions have been invaluable in improving the clarity and rigor of our manuscript. Below is our point-by-point response to the comments.

**Specific comments**

1. Line 14: The authors describe the high-resolution radiosonde dataset as providing 'near-global coverage.' However, this characterization is misleading. The distribution of stations is heavily concentrated in North America (particularly the United States) and Europe, with far fewer stations in Africa, South America, Australia, and large parts of Asia. There is also no coverage over the oceans. This uneven spatial distribution means the dataset cannot reasonably be described as 'global' or 'near-global.' I recommend revising this phrasing to more accurately reflect the actual coverage.

Response: We thank the reviewer for this insightful comment. We agree that describing the dataset as having 'near-global coverage' is an overstatement given the uneven distribution of stations, with clear gaps over the oceans and several continents. As suggested, we have revised the text to more accurately reflect the actual spatial coverage of the dataset.

Line 16: "The high-resolution radiosonde data, accumulated from 2000 to 2023 from a globally distributed yet sparse network, offers valuable insights into climatological tropopause variability."

2. Line 34: The phrase 'so-called very short-lived substances' could be reconsidered.

The tropopause may not be especially relevant for the category of very short-lived

species; just 'short-lived' may be more appropriate here. Also, the qualifier 'so-called' seems unnecessary and could be removed.

Response: Amended as suggested.

Line 35: "water vapor and short-lived substances..."

3. Line 69: The manuscript refers to ERA-Interim as a 'modern' reanalysis dataset. However, ERA-Interim was introduced nearly two decades ago and has since been replaced by ERA5 as the state-of-the-art product. Please rephrase, as calling ERA-Interim 'modern' is outdated.

Response: Thank you for this advice, there is no need to mention the ERA-Interim.

Line 71: "Modern reanalysis datasets, such as the state-of-the-art ERA5 developed by the European Centre for Medium-Range Weather Forecasts (ECMWF), and Modern-Era Retrospective analysis for Research and Applications…"

4. Line 88: I wonder if it is accurate to generalize that IGRA radiosonde data have a vertical resolution of approximately 300–400 m. IGRA typically includes measurements at standard mandatory pressure levels plus significant levels, which are reported when notable deviations in lapse rate occur. This means the vertical resolution varies substantially between soundings and over time. Please verify this with IGRA documentation and clarify.

Response: We sincerely thank the reviewer for this correct and important point. We agree that attributing a fixed vertical resolution to the IGRA dataset is an oversimplification, as its resolution is indeed highly variable due to the inclusion of standard and significant levels. We have thoroughly revised the description in the manuscript to accurately reflect the variable nature of IGRA's vertical resolution, as suggested.

Line 90: "In contrast to the original high-resolution soundings, IGRA provides a

consolidated dataset with a coarser and non-uniform vertical resolution, as it reports values at standard and significant levels (Durre et al., 2006, 2018)..."

5. Line 117: Please consider rewording 'Globally, radiosondes are launched...' for consistency with the earlier comment on line 14.

Response: We deleted this imprecise term and revised this sentence.

Line 121: "Radiosondes are launched from approximately 800 sites worldwide, regularly twice a day (Ingleby et al., 2016; Durre et al., 2018)."

6. Section 2.1: The information on radiosondes and ERA5 is somewhat mixed together. I suggest splitting this into two subsections for clarity. In addition, please provide more technical details on ERA5 (e.g., hourly temporal resolution, horizontal resolution, vertical resolution).

**Response:**

Split Content into Subsections and Reordered: Clearly separated the mixed content into distinct subsections for Radiosonde Data and ERA5 Reanalysis Data, grouping relevant details under each heading.

Added Detailed ERA5 Technical Specifications: Incorporated specific technical details for ERA5, including its hourly temporal resolution, ~31 km horizontal resolution, and 137 vertical pressure levels, and rephrased related descriptions.

Refined Data Description and Logic Flow: Streamlined the descriptions of data sources, processing, and quality control for better clarity and logical progression within each subsection.

7. Section 2.1: Please clarify whether the high-resolution radiosonde data used here are assimilated into ERA5. This is very likely the case, and if so, the datasets are not independent. Thus, this study cannot be considered a 'validation' of ERA5; it

should be framed as an 'evaluation' or 'intercomparison' (as it is already properly reflected in the title). Please make this distinction explicit.

Response: The radiosonde data integrated into ERA5 are based on standard pressure levels with lower resolution, and ERA5 does utilize a downsampled version of the high resolution radiosonde observations (Ingleby, 2017).

Although high-vertical-resolution radiosonde data are part of the assimilation process in established reanalysis data products, it's still provide a good opportunity to quantify uncertainties in the lapse rate tropopause determination from reanalysis data (Hoffmann and Spang, 2022).

Ingleby, B.: An assessment of different radiosonde types 2015/2016, Technical memorandum, https://www.ecmwf.int/en/elibrary/80268-assessment-different-radiosonde-types-20152016, 2017. Hoffmann, L., and Spang, R.: An assessment of tropopause characteristics of the ERA5 and ERA–Interim meteorological reanalyses, J. Atmos. Chem. Phys., 22, 4019–4046, https://doi.org/10.5194/acp-22-4019-2022, 2022.

8. Line 137: Why was cubic spline interpolation chosen to resample the radiosonde data from their original 5–10 m spacing to a uniform 10 m grid? At such fine spacing, cubic splines can introduce oscillations. (This question is likely not too relevant for the present study, since the authors used derived data, but would be nice if they could clarify the rationale.)

Response: We thank the reviewer for raising this valid point regarding the choice of interpolation method. The reviewer is correct that cubic splines can, in theory, introduce oscillations when interpolating to very fine scales. We chose this method primarily to generate smooth profiles that are well-suited for subsequent analysis of the atmospheric boundary layer's structure.

To clarify this for the reader, we have added a brief rationale in the manuscript on page 4, line 136. The added text states:

"This interpolation method provides a good balance between smoothness and

accuracy at the scale of interest. Given that the resampling grid spacing (10 m) is comparable to the original data spacing (5-10 m), the potential for spurious oscillations is minimal."

9. Figure 2: It may be informative to show not only the nearest ERA5 profile but those from the four surrounding grid points. This would illustrate local variability in temperature and tropopause structure and show how representative the nearest grid point is.

Response: We thank the reviewer for this excellent suggestion. We agree that comparing the radiosonde profile not only with the nearest ERA5 grid point but also with its surrounding profiles provides a more comprehensive view of the local variability and representativeness.

Following the reviewer's advice, we have modified Figure 2 to include temperature profiles from the four surrounding ERA5 grid points (in addition to the nearest one). These additional profiles are now plotted as thin orange dash lines, while the nearest ERA5 profile is highlighted with the original thick line for clear comparison with the radiosonde observation.

10. Figure 3: The scatterplot shows some very large outliers. For example, ERA5 tropopause heights at 15–17 km (typical for the tropics) relate to radiosonde values as low as 5–6 km. Please comment on these extreme discrepancies. Do they reflect limitations of the WMO definition applied to high-resolution profiles, local inversions, or possible data issues?

Response: We thank this critical observation. We agree that explaining these extreme outliers is essential for a comprehensive understanding of the data.

As suggested, we have added some text.

Line 206: "Three potential causes could lead to these outliers. First, the WMO lapserate definition is highly sensitive to fine-scale structures in high-resolution data, such as strong inversions, which can be misinterpreted as a tropopause. Second, pronounced inversions associated with phenomena like large-scale subsidence or strong fronts may genuinely satisfy the tropopause criteria at a lower altitude. Third, despite quality control, subtle data issues cannot be entirely ruled out..."

We also note that these outliers are rare events and their impact on the bulk statistical metrics presented in this study (e.g., mean bias, correlation coefficient) is negligible. Future work could focus on developing more robust tropopause identification algorithms for high-resolution profiles that are less susceptible to fine-scale noise.

11. Lines 189–194: The authors state that the mean difference (bias) in TH improves over time while the mean absolute difference (MAD) remains roughly constant. However, Table 1 shows a clear transition around 2006, coinciding with the introduction of COSMIC GPS-RO assimilation in the ECMWF reanalyses. After 2006, the radiosonde–ERA5 bias decreases, but MAD increases from ~250 m to ~350 m. This suggests that assimilation of GPS-RO data reduced the bias but increased the spread of differences, disrupting time series homogeneity. Please revisit and discuss this interpretation.

Response: We are grateful to the reviewer for pointing out this crucial information regarding the documented cold bias in the lower stratospheric temperature in ERA5 during 2000-2006 and the existence of the ERA5.1 dataset designed to correct it. This official explanation from the ECMWF perfectly aligns with and robustly confirms our (and the reviewer's) earlier interpretation that the assimilation of GNSS-RO data in 2006 caused a regime shift.

We have now integrated this key context into our revised discussion. We explicitly mention the known ERA5 cold bias in the pre-2006 period due to suboptimal background error covariances, and cite the creation of ERA5.1 as evidence of this acknowledged issue. This allows us to frame the bias reduction post-2006 not just as a correlation, but as a direct consequence of the model system being corrected by GPS-

**RO data.**

Additionally, in response to the reviewer's overarching comment regarding time series homogeneity, we have extended our discussion to include another potential source of inhomogeneity. We observed a secondary shift in our comparison statistics around 2016. We have linked this to the fact that "The starting point for ERA5 is IFS Cy41r2, which was used in the ECMWF operational medium-range forecasting system from 8 March to 21 November 2016." This change in the underlying model cycle provides a coherent explanation for the observed statistical variations at that time.

Line 221: "Table 1 details the statistical differences for each year from 2000 to 2023, revealing a gradual increase in observation data over time. Our intercomparison statistics, detailed in Table 1, reveal that the ERA5-radiosonde TH differences are not homogeneous over time. Instead, the record is marked by two transition points linked to major updates in the ERA5 system. The most pronounced shift occurs around 2006, where the mean difference decreased and the mean absolute difference increased, this change may be due to insufficient GNSS radio occultation data prior to 2006. As a result, ERA5 exhibits a significant cold bias in its stratospheric temperature analysis for the period 2000–2006. To address this, ERA5.1 reanalysis data was specifically generated, employing background error covariance applicable to the 1979–1999 period to correct this bias (Simmons et al, 2020). A secondary, subtler shift is evident around 2016, where the mean difference increased and the mean absolute difference began to decrease, concurrent with the use of the IFS Cy41r2 model cycle as the foundation for ERA5. Changes in the core model physics and assimilation settings provide a plausible explanation for this later discontinuity. Therefore, users of ERA5 TH data should be aware that its homogeneity is influenced by these key evolutionary milestones."

12. Section 3.2: The methodology for trend analysis needs clarification. Was multivariate regression applied to account for factors such as seasonality, QBO, ENSO, and volcanic activity, or were linear fits applied directly? Simple linear fits

can be misleading, as TH variability is strongly modulated by these processes. Multivariate regression has become standard in TH trend analyses; please clarify your approach and discuss limitations if only linear fits were used.

Response: We thank the reviewer for this critical comment. In our initial manuscript, we used the Sen's slope estimator directly on the time series. While robust against outliers, this method, as the reviewer rightly points out, does not isolate the long-term trend from the influence of factors like QBO and ENSO.

In response to this comment, we have completely re-analyzed our data using a multiple regression analysis that explicitly incorporates the reviewer's feedback. This new methodology is now described in detail in a newly added Section 2.3, titled "Multivariate regression model for trend analysis".

The model is formulated as follows:

$$y(t) = a \cdot t + B_1 \cdot \text{QBO1} + B_2 \cdot \text{QBO2} + C \cdot \text{ENSO} + D \cdot \text{VOL} + \sum_{k=1}^{11} E_k \cdot \text{Month}_k$$
 (2)

Line 161: "We assessed interannual variability (Section 3.2)..."

13. Table 2: The purpose of the 'ERA5-F' zonal mean comparison is unclear. The one-to-one 'ERA5-P' comparison based on collocated profiles seems more appropriate and shows better agreement. Please clarify why ERA5-F is included, and consider emphasizing ERA5-P.

Response: Thank you for your valuable comments. We fully agree that the one-to-one 'ERA5-P' is the most direct and reliable method to evaluate the accuracy of ERA5 data, and we will emphasize this as the primary conclusion in our revised manuscript. We changed 'ERA5-P' to 'ERA5', while 'ERA5-F' remains unchanged. And as you know, sparse stations cannot fairly represent the zonal mean state of an entire latitude band. Therefore, the role of ERA5-F here is to "provide a complete, reanalysis-based background reference for the zonal band".

Line 244: "Although the primary focus of our analysis is the point-to-point ERA5 data, the ERA5 zonal mean (ERA5-F) provides a comprehensive background reference..."

Line 264: "The results show strong agreement between radiosonde and ERA5 in terms of the annual mean TH..."

14. Section 4: The findings on TH uncertainties and trends of this study should be placed more explicitly in the context of recent reanalysis studies (e.g., Xian and Homeyer, 2019; Tegtmeier et al., 2020; Hoffmann and Spang, 2022; Zou et al., 2023). This would situate the results within the broader literature and highlight the contribution of this work.

Response: We thank the reviewer for this constructive suggestion. As suggested, we have added some text.

Line 282: "Recent reanalysis studies (e.g., Xian and Homeyer, 2019; Tegtmeier et al., 2020; Hoffmann and Spang, 2022; Zou et al., 2023) have advanced our understanding of TH variability,"

Line 292: "Building upon previous research by Xian and Homeyer (2019) and Hoffmann and Spang (2022), our investigation confirms an overestimation (32 m) of TH in radiosonde compared to ERA5, a bias that falls within the range of uncertainties reported in earlier intercomparison efforts."

**Technical corrections**

1. Line 167: The term 'rightward shift' of the temperature profile is awkward; consider using 'warm bias' instead.

Response: We thank the reviewer for this constructive suggestion. We have rewritten this paragraph due to the newest Figure 2.

Line 189: "...Figure 2 presents four examples of radiosonde temperature profiles alongside the corresponding profiles from the nearest ERA5 grid point and its four adjacent points. Cases (a) and (c) show good agreement in tropopause identification. In contrast, cases (b) and (d) reveal significant discrepancies despite having matched overall temperature profiles. In these instances, the high vertical-resolution radiosonde detects a distinct inversion layer—a fine-scale structure not captured by ERA5—which results in a much lower tropopause height (TH). These cases highlight how such resolved thermal structures can complicate tropopause detection, suggesting that the existing WMO definition could be further refined."

2. Lines 176 and 180: Similarly, please rephrase 'lower-left region' and 'upper-right region' of the plots for clarity.

Response: We have rephrased 'lower-left region' and 'upper-right region' to more clearly describe the physical characteristics of the data in those parts of the scatter plots. These changes have undoubtedly improved the clarity and precision of the text.

Line 201: "Close agreement between radiosonde and ERA5 is observed for THs around 13 km, with a slight positive bias in radiosonde. A pronounced discrepancy emerges at greater heights (e.g., above 13 km), characterized by increased scatter and deviation from the 1:1 line, forming a distinctive cross-like pattern..."

---

## Author Comment (AC2)

**Response to Referee**

We thank the reviewer for your time and effort in providing such insightful and constructive feedback.

We have carefully addressed all the concerns raised. The suggestions have been invaluable in improving the clarity and rigor of our manuscript. Below is our point-by-point response to the comments.

**Specific comments**

1. Though not stated anywhere in the manuscript, it is implied from Figure 2 that ERA5 output used are those with a coarsened vertical grid (37 pressure levels?) rather than the native high-resolution vertical grid. Using the coarsened data is unfair to the model for these evaluations and likely introduces greater uncertainty and bias to the results than that intrinsic to the reanalysis. The full-resolution output should be used.

Response: We sincerely thank the reviewer for raising this critical technical point. Figure 2 has been regenerated using the 137 hybrid sigma-pressure levels data. We wish to specifically clarify that the core ERA5 data used for the comparison with radiosondes in our study were always sourced from the Hoffmann and Spang (2022) ERA5-based product, which is itself built upon the full-resolution 137 model levels. The previous use of the 37-level data was limited to the schematic demonstration in Figure 2 and has now been corrected.

2. Are the radiosonde data used for comparison independent from the ERA5 assimilation? Given the source, I would expect them to all have been assimilated and thus find the profile differences in Figure 2 to be quite shocking! I'm not sure I've seen such disagreement before, which makes me question whether the colocation in space and/or time is correct. Far more attention and discussion should be given to these issues and their implications. While it is not uncommon to find large differences in tropopause height, large differences in the temperature profiles

(beyond isolated layers here and there) almost never occur.

Response: We thank the reviewer for this insightful and critical comment.

About the assimilation, the radiosonde data integrated into ERA5 are based on standard pressure levels with lower resolution, and ERA5 does utilize a downsampled version of the high resolution radiosonde observations (Ingleby, 2017).

Although high-vertical-resolution radiosonde data are part of the assimilation process in established reanalysis data products, it's still provide a good opportunity to quantify uncertainties in the lapse rate tropopause determination from reanalysis data (Hoffmann and Spang, 2022).

We have corrected the collocation errors, and the temperature profiles now show high consistency. Despite this, a statistically significant difference in tropopause height remains. This indicates that the discrepancy is not due to gross temperature biases but may stem from the challenges of reanalysis in capturing the tropopause. The role of the tropopause determination algorithm should also be considered.

Ingleby, B.: An assessment of different radiosonde types 2015/2016, Technical memorandum, https://www.ecmwf.int/en/elibrary/80268-assessment-different-radiosonde-types-20152016, 2017.
Hoffmann, L., and Spang, R.: An assessment of tropopause characteristics of the ERA5 and ERA–Interim meteorological reanalyses, J. Atmos. Chem. Phys., 22, 4019–4046, https://doi.org/10.5194/acp-22-4019-2022, 2022.

3. Furthermore on the radiosonde record, I am surprised by the minimum spatial coverage of the data used, especially since many prior studies utilizing such have had far greater global coverage. And finally, it is not clearly communicated what the vertical resolution of the native radiosonde data is. For example, I don't know what lines 136-137 mean. Is the resolution 5-10 m? If so, what is the point of using cubic interpolation to 10 m as such would be unnecessary?

Response: Thank you for these insightful questions. Regarding the spatial coverage, we

acknowledge that our dataset is more limited than some prior studies. This is because our work specifically requires long-term records with high vertical resolution, which are not widely available across the global radiosonde network. As for the vertical resolution, the native data varies by station, typically between 5-10 meters. We applied cubic interpolation to a standardized 10-meter grid not to invent new data, but to create smooth, consistent profiles. This process is essential for our subsequent analysis of the fine-scale structure within the atmospheric boundary layer, as it reduces the potential for aliasing artifacts in our results.

4. Beyond the simple tasks of measuring absolute differences, biases, and RMSE between the radiosondes and reanalyses, several additional points in the paper which are implied to be novel have been demonstrated in prior studies. This is true, for example, for the instances of large tropopause height differences between the model and radiosondes discussed in lines 178-181.

Response: We thank the reviewer for this important comment, which helps us better position the contribution of our work. We agree that the existence of tropopause height discrepancies between reanalyses and observations has been documented in prior literature.

However, using the state-of-the-art ERA5 reanalysis and a high-resolution radiosonde dataset, we provide a systematic assessment and attribution analysis of the spatial patterns, seasonal cycle, and long-term trends of these differences on a global scale. As elaborated in our response to Comment #2, our analysis specifically emphasizes that resolving the tropopause and inversion layers within fine-scale thermal structures at high-vertical-resolution is crucial.

5. There is comparatively greater emphasis given to changes in some of the statistics over time, with an implied trend suggesting that further analysis is warranted (e.g., lines 191-194), but such variability seems easily explained by the large differences in sample sizes. Also, the seasonality of the tropopause is discussed in half a

paragraph, but nothing new is provided there.

Response: We thank the reviewer for this comment. We rewrite those sentences and add some new opinion to explain the change.

Line 221: "Table 1 details the statistical differences for each year from 2000 to 2023, revealing a gradual increase in observation data over time. Our intercomparison statistics, detailed in Table 1, reveal that the ERA5-radiosonde TH differences are not homogeneous over time. Instead, the record is marked by two transition points linked to major updates in the ERA5 system..."

6. The statistical significance testing and determination is not well explained. For example, it is not clear to the reader whether the shaded regions are significant or insignificant in Figure 6.

Response: We thank the reviewer for pointing out this lack of clarity. We have modified Figure 6 and its caption as follows to ensure the reader can immediately understand which regions exhibit statistically significant trends.

- 7. Lines 63-65 imply that COSMIC is the only source of RO data, but that is not true. Response: We thank the reviewer for this insightful comment. Amended as suggested. Line 76: "GNSS-based datasets (e.g., COSMIC and the Radio Occultation Meteorology Satellite Application Facility (ROM SAF)) provide high-density measurements with near-global coverage, making them particularly suitable for analyzing global-scale tropopause characteristics (Son et al., 2011)..."
- 8. 5 & 10 year markers on Figure 1 are indistinguishable

Response: We agree with the reviewer and thank you for pointing out the poor distinguishability of the markers in the original Figure 1. We have redesigned the legend/markers (using triangles for 10-year and circles for 5-year data points).

9. The description of Figure 7 doesn't make sense. What is meant by annual average

variation? These are just time series and diagnosed trend lines, correct?

Response: We thank the reviewer for this comment and agree that the previous description of Figure 7 was unclear and potentially misleading. We have removed the trend lines from the figure, as their presence did not align with the figure's primary intended purpose.

We have refocused the figure to clearly convey its two main objectives:

- i. To demonstrate the strong consistency in the interannual variability of the data between ERA5 and radiosonde observations.
- ii. To visually present the effective recording periods and data coverage of the radiosonde data across different latitude bands.

---

## Author Response (AR2)

**Response to Referee**

We are grateful for the reviewer's valuable time and thoughtful comments. We have diligently addressed all the raised concerns, and our detailed responses are presented as follows.

**General comments**

1. In both the revised manuscript and the author response, the authors attribute a noticeable change in tropopause heights around the year 2016 to the introduction of a new IFS model cycle (Cy41r2) in ERA5. However, this explanation is not correct. The ERA5 reanalysis was produced using a single, consistent model cycle (IFS Cy41r2) and associated data assimilation system throughout the entire 1940–present record. It is a fundamental feature of reanalysis datasets that they are generated with one frozen model and assimilation system to ensure temporal consistency. Therefore, the observed changes around 2016 cannot be attributed to a switch in model cycle within ERA5. This point should be corrected in both the text and interpretation, and alternative explanations for the 2016 shift should be discussed.

Response: We sincerely thank the reviewer for catching this critical error in our interpretation. We apologize for the misunderstanding. We have now completely removed the incorrect attribution of the 2016 shift to a change in the ERA5 model cycle from both the manuscript and the discussion. Instead, we have revised the text to discuss other potential drivers, such as the strong 2015–2016 El Niño event and its potential impact on global tropopause characteristics, while acknowledging that the exact cause of this specific shift remains an open question and warrants further investigation.

Line 235: "A secondary shift occurred around 2016, marked by an increase in the mean difference and a decrease in the mean absolute difference. While this signal could be associated with climatic events like the 2015–2016 El Niño event, the mechanisms

2. The discussion of the apparent change in tropopause characteristics around 2006 should be refined: The ERA5 data assimilation system experienced a known issue during 2000–2006, leading to a cold bias in the lower stratosphere; this was corrected in the dedicated ERA5.1 reprocessing, which replaced ERA5 for that period. The Hoffmann and Spang (2022) ERA5 tropopause height dataset used in this study indeed employs ERA5.1 for 2000–2006 and ERA5 for the remaining years. Consequently, the discontinuity found around 2006 possibly reflects the transition between these two reanalysis versions. However, the launch of COSMIC GNSS-RO satellites in 2006 introduced a massive amount of high-precision temperature profile observations into the assimilation system, providing stronger observational constraints in the upper troposphere—lower stratosphere region. This increased data availability possibly contributed to the change in mean tropopause height and variability identified in the present study. I suggest discussing this aspect.

Response: We thank the reviewer for this excellent and insightful suggestion. We fully agree that clarifying the dual causes of the 2006 discontinuity is crucial. As suggested, we have now refined the discussion in the manuscript.

Line 224: "Instead, the record is marked by two transition points. The most pronounced shift occurs around 2006, characterized by a decrease in the mean difference and an increase in the mean absolute difference. This discontinuity is likely attributable to two concurrent events. First, ERA5 exhibits a significant cold bias in its stratospheric temperature analysis for the period 2000–2006. To address this, the ERA5.1 reanalysis was produced, which applied the background error covariance from the 1979–1999 period (Simmons et al, 2020). Thus, the discontinuity around 2006 likely stems from the transition from the ERA5.1 to the ERA5 reanalysis within the dataset. Secondly, the launch of COSMIC GNSS-RO satellites in 2006 markedly increased the availability of high-precision GNSS-RO data, strengthening the observational constraints on the upper

troposphere and lower stratosphere. Together, these developments may have caused the observed changes in mean tropopause height and variability..."

**Specific comments**

lines 85–95: The authors list "COSMIC and ROM-SAF" together as examples of GNSS-RO satellite datasets. This is not strictly correct: COSMIC (and COSMIC-2) are satellite missions providing GNSS-RO measurements, whereas ROM-SAF is the EUMETSAT processing facility that generates and distributes GNSS-RO products from several missions, notably the MetOp-A/B/C GRAS instruments. The authors should revise this sentence to clearly distinguish between satellite missions (e.g., COSMIC/FORMOSAT-3, COSMIC-2, MetOp-A/B/C GRAS) and processing centres (e.g., ROM-SAF) to ensure technical accuracy.

Response: We sincerely thank the reviewer for this precise and constructive comment. We agree that distinguishing between the satellite missions and the data processing centers is crucial for technical accuracy. We have revised the sentence in the manuscript accordingly to clearly reflect this distinction.

Line 76: "GNSS-based datasets from various satellite missions (e.g., COSMIC, COSMIC-2, MetOp-A, MetOp-B, and MetOp-C) provide high-density measurements with near-global coverage. Data products from these missions, including those from the MetOp-A (MetOp-B, MetOp-C) GRAS instruments as processed and distributed by the Radio Occultation Meteorology Satellite Application Facility (ROM SAF), are particularly suitable for analyzing global-scale tropopause characteristics (Son et al., 2011)..."

2. lines 135–137: The authors describe resampling the high-resolution radiosonde profiles to a uniform 10 m grid using cubic spline interpolation. This is acceptable given the similar native resolution (5–10 m) and should not introduce significant artifacts at this scale. While linear interpolation would likely be sufficient here, cubic spline interpolation does no harm and seems appropriate for producing

smoothly gridded profiles for subsequent analysis.

Response: We thank the reviewer for this positive feedback and for acknowledging the appropriateness of our methodology. We also consider linear interpolation sufficient, but we deliberately chose cubic spline interpolation to generate smoother vertical profiles, which aids the detailed structural analysis conducted in this study.

3. line 304: The manuscript still refers to "validation" of ERA5 against high-resolution radiosonde data. Since radiosonde observations are assimilated into ERA5, these datasets are not independent, and the comparison cannot be regarded as a true validation. Please replace "validation" with "evaluation" (or "intercomparison") at this location, and ensure consistent use of terminology throughout the manuscript to avoid implying that ERA5 is being independently validated by the assimilated observations.

Response: We thank the reviewer for this important comment. As suggested, we have replaced "validation" with "intercomparison" throughout the manuscript to accurately reflect that the datasets are not independent and to maintain terminological consistency. We also made several additional modifications to avoid any implication of "validation".

Line 288: "However, systematic intercomparisons of their biases and spatiotemporal patterns against high-resolution radiosonde data have yet to be fully conducted, particularly for newer reanalyses such as ERA5..."

Line 308: "A comprehensive intercomparison with high-resolution radiosonde observations demonstrates ERA5's exceptional performance in capturing TH characteristics, including absolute values, temporal variations, and spatial correlations..."

Line 327: "But many regions lack high-resolution, continuous radiosonde observations for intercomparison..."